# Rapid Screening and Identification of Antitumor Ingredients from the Mangrove Endophytic Fungus Using an Enzyme-Immobilized Magnetic Nanoparticulate System

**DOI:** 10.3390/molecules26082255

**Published:** 2021-04-13

**Authors:** Nan Wei, Jun Zhao, Guimei Wu, Wenjuan Cao, Pei Luo, Zhifeng Zhang, Gang Chen, Lu Wen

**Affiliations:** 1School of Pharmacy, Guangdong Pharmaceutical University, Guangzhou 510006, China; weinan7789@126.com (N.W.); zhao_jun0102@163.com (J.Z.); wgm19090@163.com (G.W.); adaaa890@126.com (W.C.); 2State Key Laboratory for Quality Research in Chinese Medicine, Macau University of Science and Technology, Macau 000853, China; pluo@must.edu.mo (P.L.); zfzhang@must.edu.mo (Z.Z.); 3The Center for Drug Research and Development, Guangdong Pharmaceutical University, Guangzhou 510006, China; 4Guangdong Provincial Key Laboratory of Advanced Drug Delivery, Guangdong Pharmaceutical University, Guangzhou 510006, China; 5Guangdong Provincial Engineering Center of Topical Precise Drug Delivery System, Guangdong Pharmaceutical University, Guangzhou 510006, China

**Keywords:** magnetic nanoparticles, ligand fishing, phospholipase A_2_, endophytic fungus

## Abstract

As a consequence of recent progression in biomedicine and nanotechnology, nanoparticle-based systems have evolved as a new method with extensive applications in responsive therapy, multimodal imaging, drug delivery and natural product separation. Meanwhile, the magnetic nanoparticulate system has aroused great interest for separation and purification because of its excellent magnetic properties. Phospholipase A_2_ (PLA_2_) is a highly expressed regulator to promote the growth of various cancers and is an ideal target to treat cancers. In this study, a novel strategy based on ligand–receptor interactions to discover novel PLA_2_ inhibitors was established, in which PLA_2_-functionalized Fe_3_O_4_@PLGA-PEG-NH_2_ magnetic nanoparticles were used as a supporting material combined with high-performance liquid chromatography–mass spectrometry, aiming to accelerate the discovery of novel PLA_2_ inhibitors from natural sources such as mangrove endophytic fungi. Under the optimized ligand fishing conditions, six target compounds were ultimately fished and identified to be cyclic peptides (**1**–**3**) and sterols (**4**–**6**), which compounds **1**, **2** and **4**–**6** have well-documented cytotoxicities. Compound **3** exerted better inhibitory effect on A549 cells by experiment. In conclusion, PLA_2_-functionalized Fe_3_O_4_@PLGA-PEG-NH_2_ magnetic nanoparticles-based ligand fishing provided a feasible, selective and effective platform for the efficient screening and identification of antitumor components from natural products.

## 1. Introduction

In recent decades, nanoparticle-based systems have evolved as a new system with extensive applications in multiple biomedical applications, including the diagnosis and therapy of various cancers [1,2,3,4]. In the meanwhile, these systems have also been used for separation due to their unique and inherent advantages such as high efficiency of separation and time-saving compared to the conventional bioassay-guided fractionation methods [5,6].

To date, many compounds extracted from natural products have shown considerable antitumor activities [7], although quick extraction and separation of these compounds still remains challenging. This phenomenon has attracted substantial interest in the discovering of novel methods to isolate the active ingredients from natural products. Ligand fishing, based on the target–ligand binding, was designed to attach the targets such as enzymes, membrane proteins to the carrier material, has been recognized as a convenient and efficient way to achieve multitarget or multichannel screening of natural products, coupled with chromatography or mass spectrometry (HPLC, MS or HPLC–MS) [8,9]. Moreover, new materials used in ligand fishing such as the magnetic nanoparticulate system further enhanced targets’ separation due to their great surface area and convenient separation process [10]. Recently, emerging evidence proved that ligand fishing based on magnetic nanoparticles (MNPs) has been applied to screen antitumor compounds from natural products [11,12].

Phospholipase A_2_ (PLA_2_) has been recognized as an important drug target for the initiation and progression of certain types of cancers, including liver, lung, prostate, breast cancers and chronic myelocytic leukemia [13,14,15,16,17]. PLA_2_ can specifically cut phospholipids at the sn-2 ester bond to release free fatty acids, mainly arachidonic acid (AA) and lysophospholipids (LPLs) [18]. It mediates the growth and proliferation of cancer cells primarily by releasing AA from glycerophospholipids and the corresponding metabolites from the AA. Besides, PLA_2_ may also mediate carcinogenesis by releasing LPLs, which can be metabolized into lysophosphatidic acid (LPA) to promote cell growth [19]. Therefore, PLA_2_ or inhibition of PLA_2_ was regarded as an important target and an effective way to discover drugs against cancers.

Mangrove endophytic fungi, which grow in tropical and subtropical intertidal estuarine zones, are rich in unique and bioactive compounds with potentials as new medicinal agents [20]. Recently, many novel anticancer compounds with unique structures and skeletons have been identified from mangrove fungi [21]. In addition, PLGA is a polymer with certain degree of hydrophobicity to encapsulate Fe_3_O_4_ MNPs via hydrophobic interaction, which is a Food and Drug Administration (FDA)-approved material and considered safe because of its excellent biodegradability and biocompatibility [22]. Considering the adverse limitations of Fe_3_O_4_ MNPs [23], it is highly desirable to use PLGA-PEG-NH_2_ copolymer as a linker between Fe_3_O_4_ MNPs and enzyme. In this study, for the first time, we reported a facile method to screen antitumor compounds from the fungal extracts based on a PLA_2_-functionalized Fe_3_O_4_@PLGA-PEG-NH_2_ magnetic nanoparticles (PLA_2_-MNPs) system in combination with LC–MS technology. Amino-terminated PLGA-PEG was firstly synthesized followed by the preparation of Fe_3_O_4_@PLGA-PEG-NH_2_ MNPs with emulsion, evaporation and immobilization of PLA_2_ on the surface of Fe_3_O_4_@PLGA-PEG-NH_2_ MNPs for ligand fishing. This new strategy was validated by screening antitumor compounds from the crude extract of endophytic fungus *Pseudopithomyces* sp. 1512101. The antitumor activities of screened compounds were determined by MTT assay. As a result, six compounds were efficiently isolated and identified and fusaristatin C (**3**), one of the mentioned compounds, exhibited better inhibitory effect on A549 cells. The study confirmed the suitability of using PLA_2_-MNPs as a tool of ligand fishing to discover antitumor compounds and accelerate the discovery of new drug candidates from natural products.

## 2. Results and Discussion

### 2.1. Characterization of Fe_3_O_4_@PLGA-PEG-NH_2_ MNPs

MNPs have been widely applied in chemical and biological research because of easy surface modifications, excellent stability and convenient solid–liquid separation. At present, most MNPs are based on Fe_2_O_3_, Fe_3_O_4_ and other iron oxides. However, the surface of MNPs such as Fe_3_O_4_ does not have functional groups to be linked with enzymes or proteins. PLGA-PEG-NH_2_, a di-block copolymer often used as a drug carrier, can be easily modified with various chemical groups due to its low toxicity and high biocompatibility. Therefore, in this study, Fe_3_O_4_ MNPs were designed to be encapsulated in the PEG-PLGA-NH_2_ carriers to allow magnetic separation and further decoration.

PLGA-PEG-NH_2_ was synthesized and covered on the surface of Fe_3_O_4_ MNPs to yield Fe_3_O_4_@PLGA-PEG-NH_2_ MNPs. The interaction between Fe_3_O_4_ MNP surface and PLGA-PEG-NH_2_ was mainly based on hydrophobic interaction as both Fe_3_O_4_ MNPs and PLGA were substantially hydrophobic. The Fe_3_O_4_ MNPs tend to hide inside of the hydrophobic core of the PLGA-PEG-NH_2_ polymeric aggregates to minimize the entropy in the aqueous dispersion system and, therefore, formed Fe_3_O_4_@PLGA-PEG-NH_2_ MNPs. Using DCC/NHS coupling method, activated PLGA-COOH was reacted with excess NH_2_-PEG-NH_2_ to afford PLGA-PEG-NH_2_ copolymer (Figure 1A). The synthesized PLGA-PEG-NH_2_ was characterized by FTIR and ^1^H NMR spectroscopies. FTIR spectrum (Figure 1B) confirmed the conjugation of NH_2_-PEG-NH_2_ to PLGA-COOH, as evidenced by the vibration peaks at 1625 cm^−1^ (amide C=O) and 1578 cm^−1^ (NH bond) which were not shown in the spectrum of PLGA-COOH or NH_2_-PEG-NH_2_. The successful conjugation of PLGA-COOH and NH_2_-PEG-NH_2_ was confirmed by the ^1^H NMR spectroscopy (Figure 1C). The peaks at 1.54, 5.18, and 4.79 ppm were assigned to the CH_3_, CH_2_ and CH protons of PLGA, respectively, and that at 3.60 ppm corresponded to the CH_2_ proton of NH_2_-PEG-NH_2_ blocks. Appendix A shows the FTIR spectra of Fe_3_O_4_ MNPs, PLGA-PEG-NH_2_ and Fe_3_O_4_@PLGA-PEG-NH_2_ MNPs. The spectrum of uncoated Fe_3_O_4_ MNPs exhibited the vibration peaks of typical functional groups at 550–650 cm^−1^, which were absent from the spectrum of PLGA-PEG-NH_2_ or Fe_3_O_4_@PLGA-PEG-NH_2_ MNPs, indicating that Fe_3_O_4_ MNPs had been successfully loaded into the matrix of PLGA-PEG-NH_2_ di-block copolymers.

### 2.2. Optimization of Preparation Conditions for Fe_3_O_4_@PLGA-PEG-NH_2_ MNPs

To optimize the preparation conditions for MNPs, the amount of Fe_3_O_4_ MNPs, the volume of dichloromethane and effects of preparation methods were systematically investigated.

Fe_3_O_4_@PLGA-PEG-NH_2_ MNPs were prepared by single-emulsion (o/w) and multiple-emulsion (w/o/w) methods. As shown in Appendix A, when the magnetic field strength reached 3000 Oe, the saturated magnetization of the MNPs prepared by the single-emulsion method (0.32 emu/g) was 2.6-fold to that prepared by the multiple-emulsion method (0.12 emu/g). Thus, the MNPs prepared by the single-emulsion method showed better magnetic properties.

Moreover, the effects of the amount of Fe_3_O_4_ MNPs on the particle size, PDI and saturated magnetization of MNPs were also investigated. As the amount of Fe_3_O_4_ MNPs increased, the PDI of Fe_3_O_4_@PLGA-PEG-NH_2_ MNPs remained mostly unchanged while the particle size slightly increased (Appendix A). Appendix A shows the saturated magnetization of MNPs with different Fe_3_O_4_ contents: 1000 mg (I), 800 mg (II), 600 mg (III), 400 mg (IV) and 200 mg (V). With the increasing amount of Fe_3_O_4_ MNPs, the saturated magnetization of Fe_3_O_4_@PLGA-PEG-NH_2_ MNPs increased. The growth of the saturated magnetization gradually decelerated when the amount of Fe_3_O_4_ MNPs reached 800 mg. Therefore, 800 mg of Fe_3_O_4_ MNPs was considered as an optimized amount and used in the subsequent experiments.

In addition, the content of the free amino groups on the surface of MNPs was measured. The amino groups were determined by the ninhydrin assay with the absorbance at 420 nm measured with UV spectroscopy. A calibration curve was plotted based on the results at various PLGA-PEG-NH_2_ concentrations (1.6, 1.8, 2.0, 2.2, 2.4 mg/mL), from which the content of amino group was calculated using the following formula:A = 0.2973C − 0.1348, R^2^ = 0.9952(1)
where A represents the UV absorbance of samples, C represents the concentration of amino groups in the samples. Accordingly, when the amount of dichloromethane was 0.5 mL, the concentration of PLGA-PEG-NH_2_ on the MNPs was measured and calculated as 2.21 mg/mL.

### 2.3. Characterizations of Fe_3_O_4_@PLGA-PEG-NH_2_ MNPs

Fe_3_O_4_@PLGA-PEG-NH_2_ MNPs were characterized by TEM, VSM, particle size analyzer and zeta potential analyzer. The TEM image showed spherical morphology of the prepared MNPs (Figure 2A). Figure 2A showed that the Fe_3_O_4_@PLGA-PEG-NH_2_ MNPs were dispersed in the system with an average diameter of approximately 143 nm. The graph also showed that the Fe_3_O_4_ nanoparticles were distributed in the core of the Fe_3_O_4_@PLGA-PEG-NH_2_ MNPs with different numbers as the black dots, which may cause the particles to be heterogeneous. In addition, due to the air-drying process during the TEM sample preparation, the polymeric outer layer of the Fe_3_O_4_@PLGA-PEG-NH_2_ MNPs might shrink and formed an outer dark layer in the TEM graph. Furthermore, the nanoparticles had a coercivity of 6 Oe (Appendix A) and a remanence of 0.005 emu/g (Appendix A). The average size of Fe_3_O_4_@PLGA-PEG-NH_2_ MNPs was 141 nm, with a PDI of 0.195 (Figure 2C), demonstrating excellent dispersion of the particles. The zeta potential of Fe_3_O_4_@PLGA-PEG-NH_2_ MNPs was measured to be 17.36 mV and considered to enable the nanoparticles to be relatively stable in the dispersed system.

### 2.4. Activity Study of the Immobilized PLA_2_

Immobilization of enzyme onto the surface of MNPs has many advantages compared to the enzyme solutions [24]. Notably, immobilized enzymes are more stable and can be reused repeatedly. As far as we know, there are currently no PLA_2_-modified nanoparticulate systems reported for bioactive compounds screening. Enzymes can be immobilized on the surfaces by covalent bonding or non-covalent interactions. In this study, the covalent bonding method was selected because of the higher stability of the formed enzyme-polymer conjugates. The PLA_2_ molecules were conjugated onto the surface of Fe_3_O_4_@PLGA-PEG-NH_2_ MNPs with glutaraldehyde linkers to bridge the N-terminus of the PLA_2_ molecules and the amino groups on the surface of MNPs by robust covalent bonding.

The content of –NH_2_ groups on the surface of MNPs and PLA_2_-MNPs were measured by UV method based on the ninhydrin assay. Compared with MNPs, PLA_2_-MNPs showed a lower UV absorbance at 420 nm because the number of –NH_2_ groups decreased after the reaction of PLA_2_ with –NH_2_ located on MNPs. It confirmed that the successful formation of PLA_2_-MNPs conjugates and the immobilization of the enzyme on Fe_3_O_4_@PLGA-PEG-NH_2_ MNPs.

It is necessary to measure the activities of the PLA_2_ before and after the immobilization to ensure that the conjugated enzymes are still active. Six standard PLA_2_ concentrations (0, 1.25, 2.5, 5, 10, 20 U/L) were tested to obtain a calibration curve and a linear regression equation. The enzyme concentrations of the samples were calculated using the following formula:Y = − 0.0056X^2^ + 0.2624X + 0.1188, R^2^ = 0.9996(2)
where Y represents the OD of sample and X represents the concentration of enzyme.

Three experimental groups were set to determine the concentrations of immobilized PLA_2_: (1) free PLA_2_; (2) PBS; (3) PLA_2_-MNPs prepared with different amounts of PLA_2_. According to the commercial provider, the enzymatic activity is 1.2-fold that of its concentration. As shown in Figure 3, enzymatic activity and the binding efficiency of immobilized enzyme increased significantly when the concentrations of enzyme increased from 200 U to 600 U and ceased to increase with higher concentrations of the enzyme. Therefore, PLA_2_ immobilization was carried out with 600 U PLA_2_ in a subsequent immobilization process. When the amount of PLA_2_ was 600 U, the concentrations of free PLA_2_ and immobilized enzyme were calculated to be 1.934 U/L and 1.52 U/L, respectively, and the corresponding enzyme activity levels were 2.321 U and 1.824 U, respectively. The activity of PLA_2_-MNPs was nearly 80% of that of free enzyme. The results showed that PLA_2_ had been successfully immobilized onto Fe_3_O_4_@PLGA-PEG-NH_2_ MNPs, in which the enzymatic activity of PLA_2_ was mostly maintained and ready for use thereafter.

### 2.5. Validation of the Ligand Fishing Assay

The mixture (a) of dexamethasone, curcumin and tanshinone ⅡA, the supernatant (b) after ligand fishing, and the ligand (c) fished by PLA_2_-MNPs were obtained and analyzed by HPLC referring to 3.7. Dexamethasone was chosen as a positive control to assess the specificity of the ligand fishing using a combination of PLA_2_-MNPs and LC–MS. The HPLC chromatograms revealed that tanshinone IIA and curcumin (negative controls) did not bind to PLA_2_ enzyme as they were washed off gradually in b and did not appear in c at all. On the contrary, dexamethasone was clearly observed in elute c, which showed that dexamethasone was successfully maintained and fished out with the highest amount and confirmed the specificity of the proposed method (Figure 4).

### 2.6. Ligand Fishing and LC–MS of Ligands

The HPLC chromatograms of original extract of fungus *Pseudopithomyces* sp. 1512101 (red line) and elution fractions after ligand fishing by PLA_2_ (blue line) were compared (Figure 5). The HPLC chromatogram of eluent fractions after PLA_2_ ligand fishing was significantly simpler and clearer than that of the unfished fungal extracts, implying that considerable substances without affinities to PLA_2_ were washed off and PLA_2_-MNPs showed efficient separation of target components. The total ion chromatogram of target ligands is also shown in Figure 6.

After separation and purification, six compounds **1**–**6** (Figure 5) were obtained from the extract of fungus *Pseudopithomyces* sp. 1512101. By comparing the retention time (Rt) and MS with fished ligands, apart from the peak “*” which appeared to be a fatty acid compound with extremely low polarity and failed to be isolated and identified, six compounds were obtained and identified as ligand compounds. Their retention times and molecular ion peaks are listed in Table 1.

The structures of cyclo-(4-hydroxyl-Pro-Leu) (**1**) [25], cyclo-(Pro-Val) (**2**) [26], fusaristatin C (**3**) [27], ergosterol peroxide (**4**) [28], ergosterol (**5**) [29] and cerevisterol (**6**) [30] (Figure 7) were elucidated by comparison with previously reported NMR spectroscopy and MS data. Their ^1^H and ^13^C NMR spectroscopy data are shown in Appendix A.

Human lung carcinoma cell line A549, bone marrow neuroblastoma cell line SH-SY5Y and cervical cancer cell line HeLa were selected for cytotoxicity experiment as PLA_2_ has been recognized as an important drug target for the initiation and progression of these types of cancers. The results of PLA_2_ inhibitory assay demonstrated that compound 3 had an inhibitory effect when the concentration was 60 μM and 100 μM (Appendix A). According to previous literatures, compounds **1**, **2** and **4**–**6** have been reported with various degrees of cytotoxicity against human cancer cells (Table 2). Therefore, only compound **3** was selected to test its inhibitory activities against human cancer cells in this study. Among the in vitro tests of compound **3** against human lung carcinoma cell line A549, bone marrow neuroblastoma cell line SH-SY5Y and cervical cancer cell line HeLa, the MTT assay revealed that A549 cell line was the most sensitive to the compound **3** and its IC_50_ was calculated to be around 10.10 μM (Table 2), lower than the reported results of the other compounds, apart from the result of compound **4** against K562 cell line. The results suggested that compound **3** was a promising inhibitor for PLA_2_ and a potential lead compound worthy of further antitumor drug research.

## 3. Experimental Section

### 3.1. Materials and Chemicals

Poly(dl-lactic-*co*-glycolic acid) 50/50 (PLGA-COOH, MW ~10,000) was purchased from Shandong Medical Equipment Institute (Jinan, China). Polyethylene glycol bis-amine (NH_2_-PEG-NH_2_, MW ~2000) was obtained from Xi’ an Ruixi Biotechnology Co., Ltd. (Xi’ an, China). N-Hydroxysuccinimide (NHS), dicyclohexylcarbodiimide (DCC), polyvinyl alcohol (PVA, MW ~3100), PLA_2_ (PLA_2_ from porcine pancreas), deuterated reagent and methyl thiazolyl tetrazolium (MTT) were obtained from Sigma-Aldrich (Shanghai, China). Fe_3_O_4_ MNPs (20 nm) were bought from Shanghai Ziming Biotechnology Co., Ltd. (Shanghai, China). Glutaraldehyde solution (3 wt %) and tris(hydroxymethyl)methyl aminomethane (Tris) were provided by Aladdin Chemistry (Shanghai, China). Dexamethasone was obtained from Hubei Yuancheng Saichuang Technology Co., Ltd. (Hubei, China). Curcumin was obtained from Shanghai Maclean Biochemistry Co., Ltd. (Shanghai, China). Tanshinone IIA was obtained from Shenzhen Meihe Biological Technology Co., Ltd. (Shenzhen, China). Sodium cyanoborohydrid and 5-Fluorouracil (5-FU) were obtained from Shanghai Macklin Biochemical Co., Ltd. (Shanghai, China). PLA_2_ ELISA kit was purchased from Shanghai Lichen Biotechnology Co., Ltd. (Shanghai, China). Potato dextrose agar (PDA), glucose, peptone, yeast extract and sea salt were obtained from Guangdong Huankai Microbial Sci. & Tech. Co., Ltd. (Guangzhou, China). Methanol (MeOH) was obtained from Merck (HPLC-grade, Darmstadt, Germany). Column chromatography silica gel (200–300 mesh) was obtained from Qingdao Ocean Chemical Co., Ltd. (Qingdao, China). Sephadex LH-20 was obtained from GE Healthcare Bio-Sciences AB (Shanghai, China). DMEM and 1640 medium were obtained from Gibco (California, USA). The solvents and reagents used in this study were classified as analytical grade.

### 3.2. Fungal Strain

The fungus 1512101 was isolated from the leaves of mangrove plant Sonneratia caseolaris, which was collected in October 2015 from the Nansha Mangrove Nature Reserve in Guangzhou, China. The fungal strain was identified to be most similar (99%) to that of Pseudopithomyces sp. 1512101 (compared to MF919624.1). The voucher specimen was stored in our laboratory at 4 °C.

### 3.3. Cell Culture

A549, SH-SY5Y and HeLa cells were generously provided by Professor Mao (School of Pharmacy, Guangdong Pharmaceutical University, China). They were incubated in DMEM and 1640 medium (containing 100 U/mL penicillin, 10% (*v/v*) fetal bovine serum and 100 µg/mL streptomycin) in an incubator (5% CO_2_, 37 °C).

### 3.4. Apparatus and Characterization

UV absorptions were measured using a UV-2550 spectrophotometer (Shimadzu, Kyoto, Japan). All surface functional groups were detected in KBr using a Fourier transform infrared (FTIR) spectrometer (Perkin Elmer, Waltham, MA, USA). The morphology of nanomaterials was observed using a JEM-3100F transmission electron microscope (TEM, JEOL Co., Tokyo, Japan). The particle size of nanomaterials was measured using a Malvern Nano S90 particle size analyzer (Malvern, UK). The zeta potential was recorded on a Delsa TM Nano laser nanoparticle analyzer (Beckman Coulter, Inc., Brea, CA, USA). The magnetic properties were tested with a MPMA XL-7 vibrating sample magnetometer (VSM, Quantum Design, CA, USA). Nanoparticles were freeze-dried using an LGJ-10 freeze dryer (Beijing Songyuan Huaxing Technology Development Co., Ltd., Beijing, China). A YM75 vertical autoclave (Shanghai Sanshen Medical Equipment Co., Ltd., Shanghai, China) was used to sterilize culture medium. A SPX-250C constant temperature and humidity incubator (Shanghai Boxun Industrial Co., Ltd. Medical Equipment Factory, Shanghai, China) and TCYQ shaker (Taicang Experimental Equipment Factory, China) were used for culturing strains. The inhibitory activities of ligands were determined using a Multiskan FC microplate reader (Thermo Fisher Scientific, Waltham, MA, USA). HPLC–MS, carrying out on an Agilent 6120 LC system (Agilent Technologies, Santa Clara, CA, USA), was equipped with a 6520 quadrupole-time-of-flight mass spectrometer (Agilent Technologies, Santa Clara, CA, USA) using an ESI ion source. A Bruker Avance III 400 NMR spectrometer (Bruker, Germany) was employed to obtain the ^1^H NMR (400 MHz) and ^13^C NMR (100 MHz) spectra of the samples using TMS as an internal standard. A Micromass LCT mass spectrometer (Waters, USA) was used to measure the ESI-TOF mass spectra of the sample, and the accurate mass was determined and calibrated using a lock mass setup. Cells were cultured in an HER Acell 150i incubator (Thermo Fisher Scientific, Waltham, MA, USA).

### 3.5. Preparation of PLA_2_-MNPs

#### 3.5.1. Preparation of PLGA-PEG-NH_2_ di-block Copolymer

PLGA-PEG-NH_2_ di-block copolymer was synthesized with the formation of amide linkage between NH_2_-PEG-NH_2_ and activated PLGA-COOH as described in previous studies [37,38,39,40,41]. PLGA-COOH (100 mg, 0.01 mmol), 5 molar excesses of NHS (5.6 mg, 0.05 mmol) and DCC (10.3 mg, 0.05 mmol) were dissolved in anhydrous dichloromethane (DCM). The mixture was stirred under N_2_ at room temperature for 24 h to activate the carboxyl group of PLGA-COOH. The resulted solution was filtered to remove the formed dicyclohexylurea. The activated PLGA was added dropwise to excess amount of NH_2_-PEG-NH_2_ (100 mg, 0.05 mmol) dissolved in anhydrous DCM with gentle stirring. The stoichiometry of PLGA-COOH/NH_2_-PEG-NH_2_ was 1:5. The mixture was allowed to be reacted for another 24 h under room temperature. The DCC and NHS were removed from the concentrated solution by precipitation in ice-cold diethyl ether. The amine-terminated PEG-PLGA-NH_2_ di-block copolymer was further purified in excess-cold methanol and lyophilized to afford the final product.

#### 3.5.2. Preparation of Fe_3_O_4_@PLGA-PEG-NH_2_ MNPs

Fe_3_O_4_@PLGA-PEG-NH_2_ MNPs were prepared with a classical solvent emulsion/evaporation method [42,43]. Briefly, 800 mg of Fe_3_O_4_ MNPs and 100 mg of PLGA-PEG-NH_2_ were suspended in 0.5 mL of dichloromethane by vortex, giving an oil phase that was then added to 2 mL of 3 wt % aqueous PVA solution and sonicated for 2 min. The emulsion was subsequently added to 15 mL of 1 wt % PVA solution, and the organic phase was removed through reduced pressure evaporation. The dispersion was centrifuged at 6000 rpm for 20 min to remove the unencapsulated Fe_3_O_4_ MNPs. The prepared Fe_3_O_4_@PLGA-PEG-NH_2_ MNPs product was then freeze-dried and stored at 4 °C.

During the synthesis of Fe_3_O_4_@PLGA-PEG-NH_2_ MNPs, the effects of different reaction conditions on the magnetic properties of MNPs, including preparation methods and amounts of Fe_3_O_4_ MNPs, were assessed.

#### 3.5.3. Preparation of PLA_2_-MNPs

PLA_2_ was bound onto the surface of Fe_3_O_4_@PLGA-PEG-NH_2_ MNPs in the form of Schiff base linked by glutaraldehyde. Briefly, 16 mg of Fe_3_O_4_@PLGA-PEG-NH_2_ MNPs were suspended in PBS (0.01 M, pH 7.7) and vortexed for 10 min. Subsequently, 20 mg of sodium cyanoborohydride and 1 mL of 3 wt % glutaraldehyde were added to the solution and oscillated at 37 °C for 3 h. Afterwards, the aldehyde-activated MNPs were magnetically separated from the supernatant using a magnet and washed twice with PBS to remove excess glutaraldehyde. Then, the activated Fe_3_O_4_@PLGA-PEG-NH_2_ MNPs linked with glutaraldehyde were added to PBS containing 600 U of PLA_2_. The solution was vibrated at 37 °C for 2 days. After immobilization, the PLA_2_-immobilized nanoparticles were rinsed twice with water, redispersed in Tris-HCl buffer (1 M, pH 8) and stored at 4 °C prior to use. During the preparation of PLA_2_-MNPs, the effects of various amounts of PLA_2_ on the enzymatic activity of MNPs were also assessed.

### 3.6. Enzyme Activity Assay

The activity of PLA_2_ was tested using an ELISA kit. Enzymatic reactions were performed on a 96-well microplate. Free PLA_2_ and immobilized PLA_2_ were dissolved in PBS (0.01 M, pH 7.7). PLA_2_ was preincubated with HRP-conjugate reagent at 37 °C for 1 h, followed by addition of substrate chromogen solution A (trimethoxybenzaldehyde) and chromogen solution B (tetramethylbenzidine), and incubation at 37 °C for 15 min till the stopping buffer was added into each well. The optical density (OD) of the sample was measured using a microplate reader (450 nm). A standard curve of the samples was plotted using the averaged OD values of six standard concentrations. Finally, the concentration of PLA_2_ was determined by comparing OD value of the sample to the standard curve.

### 3.7. Establishment and Validation of Ligand Fishing Assay

Dexamethasone, a typical PLA_2_ inhibitor, was chosen as the positive control for the verification of the proposed method. Dexamethasone, curcumin and tanshinone IIA (2 mM each, equimolar) were mixed in PBS as a model sample (a). Three mL of the model sample were added to PLA_2_-MNPs dispersion and incubated for 3 h at 50 °C. After magnetic separation, the supernatant (b) was collected and the MNPs were washed 3 times with PBS. One mL of methanol was used to redisperse the PLA_2_-MNPs for 1 h to dissociate bound components. After separation, the supernatant (c), the mixture a and the supernatant b were collected and analyzed by HPLC after filtration. The column temperature was set at 25 °C. The eluent flow rate was 0.5 mL/min. The mobile phase consisted of solvent A (methanol) and solvent B (0.1%, *v/v*, formic acid/water). The gradient elution program was optimized as below: 50–85% A at 0–10 min, 85% A at 10–25 min, 85–90% A at 25–30 min, 90% A at 30–40 min. The sample injection volume was 20.0 μL.

### 3.8. Fishing Potential Ligands from Fungal Extract

#### 3.8.1. Preparation of Fungal Extract

The fresh mycelia of *Pseudopithomyces* sp. 1512101 were grown on PDA medium at 28 °C for 3–4 days and then inoculated into conical flasks (500 mL) containing 250 mL of PYG medium (10 g/L glucose, 2 g/L peptone, 1 g/L yeast extract, 2 g/L sea salt, pH 6.0–7.0). After incubation for 3–4 days at 28 °C on a rotary shaker at 120 rpm, 5 mL of the culture medium was transferred as the seed into 500 mL flasks containing PYG medium (250 mL). The conical flasks were then incubated for a month at room temperature (25–30 °C). Subsequently, the whole fermented cultures (150 L) were filtered through cheesecloth. The mycelia were separated to obtain a culture broth. Ethyl acetate was added to the culture broth for extraction until the upper layer became colorless, while the mycelia were extracted 3 times with methanol and concentrated. Subsequently, the concentrate was further extracted with ethyl acetate and concentrated to obtain the crude extract (38.4 g).

#### 3.8.2. Application of Ligand Fishing in Fungal Extract

The above fungal extract (3 mL, 1 mg/mL) was incubated at 50 °C with PLA_2_-MNPs for 3 h. After the separation process, PLA_2_-MNPs were taken out and washed 3 times with PBS to remove the components with no or low affinity to PLA_2_ followed by incubation in methanol (1 mL) for 1 h to dissociate the specifically bound components. The methanol solution containing potential ligands was collected.

### 3.9. Analysis of Ligands by LC–MS

LC–MS technology was used to analyze the obtained ligands. A Phenomenex C18 column (250 mm × 4.6 mm, 4.5 μm) was used for chromatographic separation. The column temperature was maintained at 25 °C. The eluent flow rate was set at 1 mL/min, and the mobile phase consisted of solvent A (methanol) and solvent B (0.1%, *v/v*, formic acid/water). The gradient elution program was optimized as below: 50% A at 0–5 min, 50–90% A at 5–45 min, 90–100% A at 45–60 min and 100% A at 60–70 min. The injection volume of the sample was set as 20.0 μL. MS data were acquired in the positive/negative ion mode by ESI and analyzed by PeakView 1.2^®^ Software.

### 3.10. Isolation of Ligands

Next, 38.4 g of extract were separated on a silica gel column with gradient elution by using petroleum ether/ethyl acetate (90:10, 70:30, 50:50, 30:70, 0:100, *v/v*) and ethyl acetate/methanol (50:50, 0:100, *v/v*) to yield 15 fractions (Frs. 1–15). The retention time of the contents in each fraction was compared with that of ligands fished out under the same HPLC conditions. Frs. 4–9 were further separated by column chromatography including silica gel, preparative chromatography and Sephadex LH-20 column, giving target compounds **1**–**6**.

### 3.11. In Vitro Cytotoxicity Assay

Compound 3 was evaluated for its inhibitory activities against PLA_2_ using a sPLA_2_ inhibitor screening assay kit firstly. Then, the cytotoxicities of compound 3 against human lung carcinoma cell line A549, bone marrow neuroblastoma cell line SH-SY5Y and cervical cancer cell line HeLa were determined by the MTT assay. The cells were subcultured when 80–90% confluence was reached, seeded in a 96-well microplate at the density of 5 × 104/mL per well and incubated with various concentrations of compound 3 for 24 h. After, 5-FU was used as the positive control. The cells were treated with 10 µL of MTT in PBS (5 mg/mL) and then incubated for another 4 h. The cells were dissolved in 100 μL of dimethyl sulfoxide. Finally, the OD of the sample at 490 nm was measured and recorded using a microplate reader.

## 4. Conclusions

In summary, a new strategy using ligand fishing based on PLA_2_-MNPs with LC–MS was established to separate and analyze bioactive components from natural products. Using this strategy, six ligands of PLA_2_ were rapidly extracted and identified from mangrove endophytic fungi. Furthermore, the cytotoxicity of these compounds was also evaluated. This robust and relatively convenient ligand fishing approach using PLA_2_-MNPs can be applied to screen more antitumor ingredients from natural extracts. In the future, we can construct a new nanoparticle system to load targeted enzyme compounds for the treatment of cancer in vivo.

## Figures and Tables

**Figure 1 molecules-26-02255-f001:**
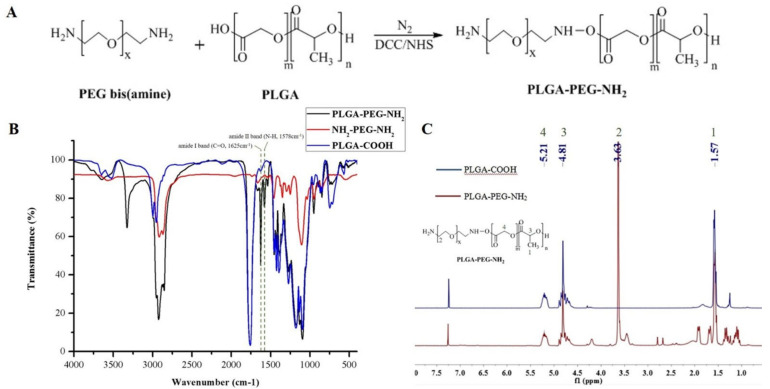
Characterizations of synthesized copolymers by ^1^H NMR and FTIR spectroscopies. (**A**) Synthesis scheme; (**B**) FTIR spectra of PLGA-COOH, NH_2_-PEG-NH_2_ and PLGA-PEG-NH_2_; (**C**) ^1^H NMR spectra of PLGA-COOH and PLGA-PEG-NH_2_.

**Figure 2 molecules-26-02255-f002:**
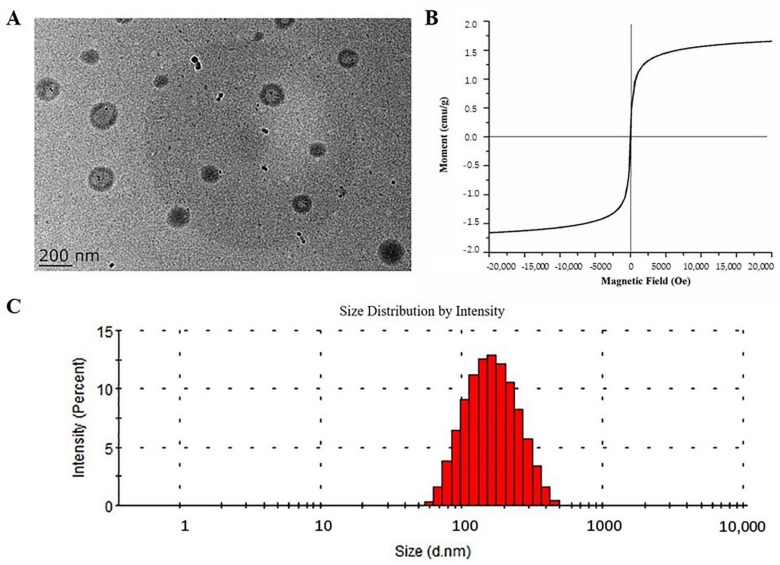
Characterizations of Fe_3_O_4_@PLGA-PEG-NH_2_ MNPs. (**A**) TEM image; (**B**) hysteresis loop; (**C**) particle size distribution histogram.

**Figure 3 molecules-26-02255-f003:**
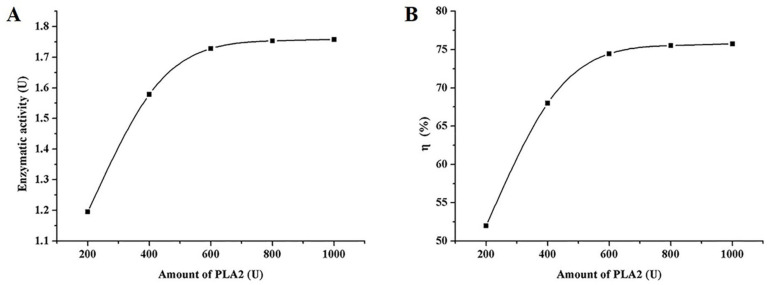
Influence of amounts of PLA_2_ on (**A**) enzymatic activity and (**B**) the binding efficiency of immobilized enzyme.

**Figure 4 molecules-26-02255-f004:**
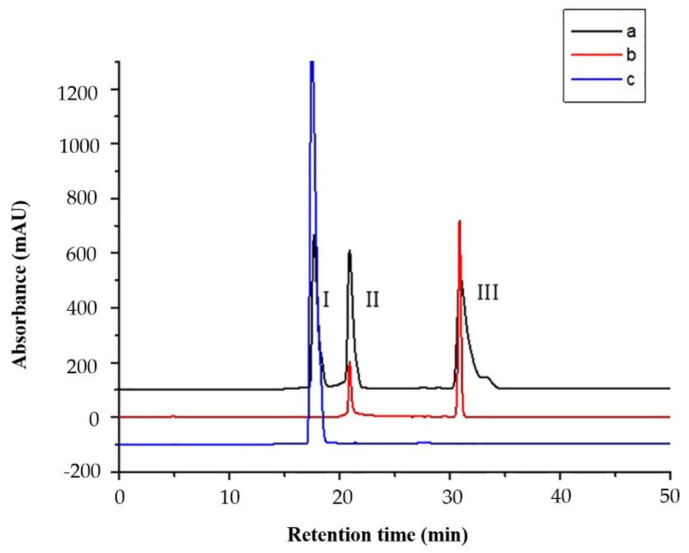
HPLC chromatograms (254 nm) of (**a**) the ligand fishing experiment with a prepared test mixture consisting of dexamethasone (I), tanshinone IIA (II) and curcumin (III). Non-binders II and III were washed out (**b**), whereas binder I was eluted with methanol (**c**).

**Figure 5 molecules-26-02255-f005:**
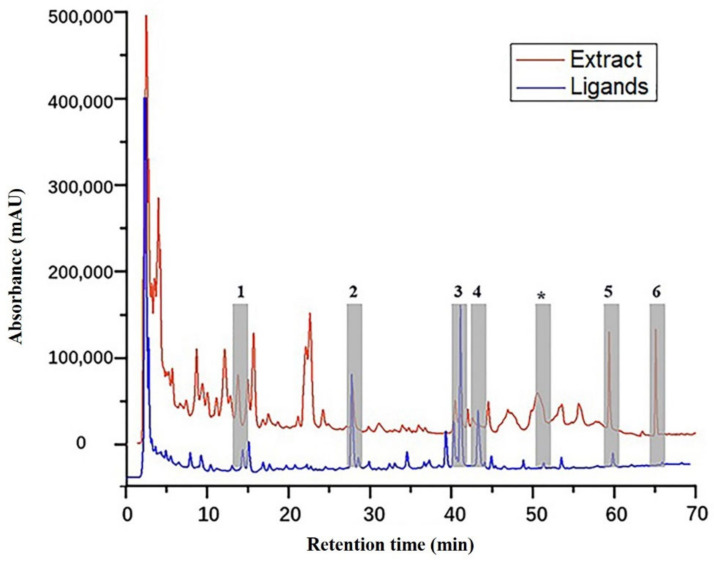
HPLC chromatograms of the ligands and extract.

**Figure 6 molecules-26-02255-f006:**
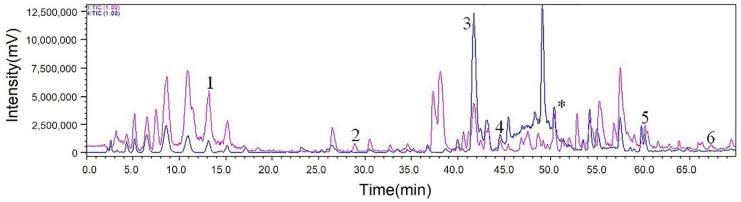
Total ion chromatograms of ligands (positive ion mode: red line; negative ion mode: blue line). The peak “*” which appeared to be a fatty acid compound.2.7. Identification of Ligands and Evaluation of Their Cytotoxicity.

**Figure 7 molecules-26-02255-f007:**
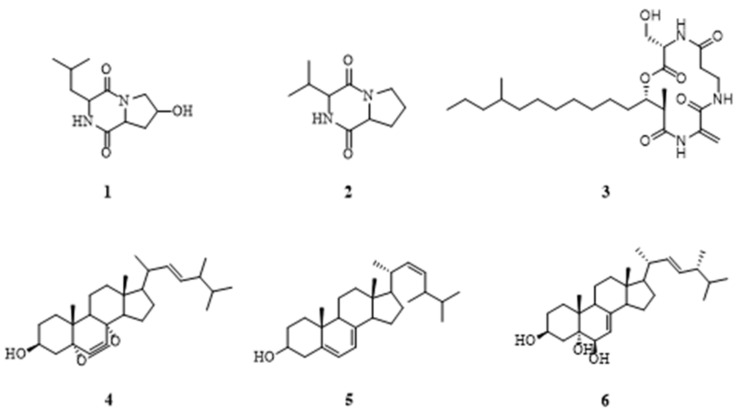
Chemical structures of the isolated ligands.

**Table 1 molecules-26-02255-t001:** LC–MS data of ligands found in fungal extracts.

Compound	Rt (min)	Formula	+ESI, m/z	−ESI, m/z
**1**	13.124	C_11_H_18_N_2_O_3_	453.3417	487.3021
**2**	28.860	C_10_H_16_N_2_O_2_	219.1898	-
**3**	41.381	C_25_H_43_N_3_O_6_	482.3256, 504.3039	516.2833
**4**	44.528	C_28_H_44_O_3_	451.3171	463.2874
**5**	60.197	C_28_H_44_O	397.4140	431.3750
**6**	67.295	C_28_H_46_O_3_	-	429.2816, 465.2551

**Table 2 molecules-26-02255-t002:** Effects of ligand compounds on tumor cells.

Ligand Compound	Cytotoxicity
**Compound 1**	Human chronic myelogenous leukemia cell line K562 (367.38 μM, 33.3%) [31].
**Compound 2**	Mouse breast cancer cell line tsFT210 (25.35 μM, (13.3 ± 1.6%) [32]. Liver cancer cell line HepG2 (253.55 μM, 17%) [33]. Prostate cancer cell line LNCaP (253.55 μM, 53%) [33].
**Compound 3**	Human lung carcinoma cell line A549 (IC_50_ = 10.10 μM).
**Compound 4**	Human chronic myelogenous leukemia cell line K562 (IC_50_ = 4.30 μM) [34]. Cervical cancer cell line HeLa (IC_50_ = 15.76 μM) [35].
**Compound 5**	Human chronic myelogenous leukemia cell line K562 (IC_50_ = 40.17 μM) [34]. Breast cancer cell line MCF-7 (IC_50_ = 112.65 μM) [36].
**Compound 6**	Human chronic myelogenous leukemia cell line K562 (IC_50_ = 169.68 μM) [34]. Breast cancer cell line MCF-7 (IC_50_ = 429.03 μM) [36].

## Data Availability

The data presented in this study are available on request from the corresponding author.

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
