# Peer review of "Rapid Screening and Identification of Antitumor Ingredients from the Mangrove Endophytic Fungus Using an Enzyme-Immobilized Magnetic Nanoparticulate System"

_molecules, 2021, doi:10.3390/molecules26082255_

Round 1
Reviewer 1 Report
The article “Rapid screening and identification of antitumor ingredients from the mangrove endophytic fungus using an enzyme-immobilized magnetic nanoparticulate system” by Nan Wei et al. presents a profound study on the use of magnetic nanoparticles for the screening of naturally occurring compounds for their anti-tumor activity. The authors synthesize nanoparticles based on PLGA-PEG diblock-copolymer with embedded Fe3O4 nanoparticles which form a stable aqueous dispersion. Subsequently, the enzyme phospholipase A2 is immobilized on the particle surface by covalent bonding. These magnetically labelled enzymatically active particles are then used for the “fishing” of compounds which are capable to bind to phospholipase A2 and which possibly block this enzyme, a property linked to anti-tumor activity.
The study deals with the whole range from the synthesis of the nanoparticles up to the identification of the possible anti-tumor agents, therefore it covers a wide range of topics. Consequently, it may be interesting to readers working in very different fields and, to my opinion, is well suitable for publication in MOLECULES.
However, there are some issues which should be addressed or corrected by the authors in order to improve the manuscript prior to publication:
- On page 4 in line 150, the authors state that the stoichiometry between the PLGA and the PEG components is 1:5. However, given that the PLGA component has an MW of 10 000 g/mol and the PEG component having one of 3000 g/mol, and that both are being used on a 100 mg scale, one would rather expect a stoichiometry of 1:3. Could the authors explain for this discrepancy?
- On page 8 in line 304, the authors quantify the concentration of free NH2-groups to 2.21 mg/mL. This statement does not make sense, as it is not relevant for the NH2-group itself, but rather refers to the corresponding concentration of PLGA-PEG-NH2. The authors should either make this clear, or they should calculate the molarity of free NH2-groups instead.
- In Figure 2A, the authors show micrographs of their nanocarriers, stating a “uniformly spherical morphology”. From the structure of the particles, one would expect the Fe3O4 nanoparticles to show up, as their size of approximately 20 nm should be above the resolution limit. Can they be detected by TEM? How do they distribute inside the particles? From Fig. 2A, one may have the impression that the outer layer of the particles appears darker than the core. Is that true? There should be some more discussion about the structure of the particles, as they are expected to be distinctly heterogeneous.
- In section 3.5 on pages 9 and 10, it would be helpful to repeat the meaning of the steps a, b and c. It is explained in section 2.7, however, it would improve the readability if it would be mentioned in 3.5 again.
- Table 1: A safe identification of the compounds 1 to 6 would include a check on the retention time using a reference compound. Has this been done?
- Table 2: the efficiencies of compounds 1 over 6 regarding their anti-tumor activities is given occasionally in µg/mL, occasionally in µM. The comparison would be much easier if the same unit would be used in each case, preferably µM.
With these small additions and corrections, I would support the acceptance of the article.
Reviewer 2 Report
Review on manuscript
Rapid Screening and Identification of Antitumor Ingredients from the Mangrove Endophytic Fungus Using an Enzyme-Immobilized Magnetic Nanoparticulate System
The key subject of the present manuscript is to identify new, natural small molecule PLA2 inhibitors, by LC-MS method using optimized ligand fishing by the means of enzyme-immobilized magnetic nanoparticles. Phospholipase A2 (PLA2) enzyme cleave phospholipids preferentially at the sn-2 position, liberating free fatty acids and lysophospholipids. It is assumed that PLA2 can serve as pharmacological targets for the therapeutic treatment of several diseases, including cardiovascular diseases, atherosclerosis, immune disorders and cancer. To inhibit PLA2 activity, there are several pharmacological way, one of them are the use of small molecule PLA2 inhibitors.
The authors designed a nanoparticle-based strategy to screen and identify natural molecules with PLA2 inhibitor properties from the extract of fungus 1512101 isolated from the leaves of mangrove plant.
At the beginning, I found the idea and the experiment design very impressive. However, after checking the references, I found a very similar paper of Li et al: Journal of Chromatography B, 960 (2014) 166–173, entitled “Rapid screening and identification of a-amylase inhibitors from Garcinia xanthochymus using enzyme-immobilized magnetic nanoparticles coupled with HPLC and MS”. Not only the title but the basic idea and the whole structure of the study is very similar. The differences consist in the target enzyme: PLA2 instead of a-amylase, and a little modified nanoparticle assembly is prepared: MNP@PLGA-PEG-NH2 instead of silica-coated MNP (using TEOS) submitted further to amination (with APTES) an fluorescent labelling (FITC).
What is the benefit of using PLGA-PEG-NH copolymer as a linker between MNP and enzyme? Would not be enough to use NH2-PEG-NH2 without PLGA or PLGA plays an extra role?
The preparation and characterization of the nanoparticle system is well presented, however, some minor question arises:
- Preparation of PLGA-PEG-NH copolymer: the characterization of the product was performed by FTIR and 1H NMR spectroscopies. I suggest for a better visualization to present IR spectra in absorbance and a possible vertical shift of the spectra facilitate the understanding. I also suggest to mark the newly formed amide I and amide II bands upon DCC/NHS coupling. Are the IR spectra measured in KBr pellet?
- 2: In the TEM picture it seems that beside MNP@PLGA-PEG-NH2 particles with average diameter of 150 nm, small unreacted MNPs with cca. 20 nm are also present. Was any separation procedure applied? The DLS figure is oversized, and, based on TEM picture, inaccurate (and not histogram). It seems unbelievable a polydispersity index (PDI) of 0.2 which suggest a monodisperse system.
- What is the driving force between Fe2O3 MNP surface and PLGA-PEG-NH2 interaction? There is no explanation or argumentation.
- To prove the immobilization of PLA2 towards MNP UV measurements were performed. I guess that a ninhydrine test was applied. The paragraph, however, between rows 328-332 is not formulated clearly. A more precise and detailed discussion of the result is needed.
The main part of the results dealing with ligand fishing is only tangential, however, is the center topic of the study. About ligand fishing validation: from Fig4 is clear that the dexamethasone was bound, while the non-binders tanshinone IIA and curcumin were washed. Can we know something about binding efficacy of dexamethasone or about equilibrium conditions? How the concentration can affect the binding constant?
The description of the identification of ligands is unsatisfactory. I guess more argumentation is needed to convince the readers.
The cytotoxicity is sketched. The results are not presented. What was the reason for choosing the given cell lines? How this correlates with data from literature. Is the Compound 3 promising inhibitor for PLA2? How the outcomes of cytotoxicity study correlates with the performance of the other Compounds? There is no discussion. How should Table 2 interpret? At least the IC50 values might be compared.
The Conclusion is very meaningless. What the sentence in row 395 “Futhermore, these compounds we separated can be structurally modified and immobilized on magnetic nanoparticles for drug delivery targeting PLA2.” means? The same for the last sentence in row 407.
There is nowhere highlighted the novelty of the work and the presented nanoparticulate system why is more beneficial compared to that from studies already published.
As a general opinion, the work is interesting, however far not original. Nanoparticle system preparation and characterization is well described, some minor revision and implementation needed. Ligand fishing, LC-MS results and cytotoxicity assays are sketchy and not acceptable. More detailed analysis and argumentation are needed.
In the present form the manuscript is not ready for publication. One of the main issue is the lack of novelty of the work. At least some arguments about the benefit of the novel nanoparticulate system are required. The ligand fishing experiments also miss the originality, are incomplete with insufficient discussion. A strong re-thinking of the whole concept of the work with some additional measurements and much more detailed discussion is needed.
Round 2
Reviewer 2 Report
Review on revised manuscript
The authors augmented significantly the manuscript and answered adequate all the questions and comments. However, I suggest to complete also the introduction part by adding same detailes, e.g. why the authors have choosen PLGA as linker for enzyme-immobilization.
One question remained: what is marked in FigS1 presenting IR spectra.
After these minor revisions I recommend the manuscript ot be published in Molecules.
